# Digital Core Permeability Computation by Image Processing Techniques

Qinzhuo Liao [1] , Shaohua You [2],* , Maolei Cui [1] , Xiaoxi Guo [3] , Murtada Saleh Aljawad [4] and Shirish Patil [4]

1 State Key Laboratory of Shale Oil and Gas Enrichment Mechanisms and Effective Development, SINOPEC Petroleum Exploration and Production Research Institute, Beijing 100029, China; liaoqz@cup.edu.cn (Q.L.); cuimaolei@163.com (M.C.)
2 College of Petroleum Engineering, China University of Petroleum-Beijing, Beijing 102249, China
3 State Grid Information & Telecommunication Branch, Beijing 100761, China; guoxiaoxi.bj@foxmail.com
4 College of Petroleum Engineering & Geosciences, King Fahd University of Petroleum and Minerals, Dhahran 31261, Saudi Arabia; mjawad@kfupm.edu.sa (M.S.A.); patil@kfupm.edu.sa (S.P.)
* Correspondence: youshaohua01@163.com

**Abstract:** Calculation of REV (representative elementary volume) properties of geological porous media refers to the process of creating a 3D digital representation of a rock sample, typically obtained from imaging techniques such as X-ray microtomography. This technique allows for a detailed analysis of the internal structure and the properties of rocks, as well as precise calculation of various flow parameters. However, one major challenge with calculation of REV properties of geological porous media is the high computational cost required to generate accurate results, especially for large and complex samples. In this study, we constructed 3D digital cores of dune sand and fractured shale using CT scanning technology, and then used two image processing techniques, namely digital core image resampling and cutting, to reduce the computational cost of calculating digital core permeability. Next, a fast permeability calculation method is employed to reduce the complexity of permeability calculation. Finally, we summarized the applicability of different image processing methods to different rock samples, and provided prerequisites for high computational cost digital core permeability calculation.

**Keywords:** permeability; digital core; image processing; shale; dune sand

## 1. Introduction

Calculation of REV (representative elementary volume) properties of geological porous media is a rapidly evolving field in petroleum geology. It utilizes advanced imaging technologies and computational methods to analyze the properties of rock samples without the need for physical sampling. By creating high-resolution images of rock samples, digital core analysis enables researchers to study the structure of rocks, pore network, mineral composition, and other properties. These properties can provide valuable insights into reservoir characteristics and production optimization.

In recent years, the rapid development of digital core technology can be attributed to the continuous advancement of CT scanning technology and the widespread utilization of advanced equipment such as Micro-CT, Nano-CT, and FIB-SEM. This technology has been applied to the petrophysical analysis of various rock types. In general, cracks, pore throats, or different mineral components in rocks will illustrate different grayscale values after scanning. Therefore, the grayscale images obtained from rock core scanning can be applied to a 3D digital core model, which reflects the real fracture and pore structure of the rock core. Based on this model, petrophysical analysis, elastoplastic deformation simulation, microscale flow simulation, thermal conductivity simulation, and other types of simulation analysis can be performed on the rock core. Currently, there are two main methods for constructing digital core samples: numerical reconstruction and physical

experiment. The numerical reconstruction method can be further divided into the random method and the process simulation method. The random methods reconstructs the 3D digital core samples by statistical analysis of the geometric properties of pore spaces based on the 2D images of the rock samples including the Gaussian random field method, the simulated annealing method, the sequential indicator simulation algorithm, and the geological statistical method [1]. The process simulation method reconstructs the 3D digital core samples by simulating the geological formation process of the rock including sedimentation, compaction, cementation, and diagenesis [2–4]. Although the cost for the numerical reconstruction method is lower, it is prone to significant bias and cannot reflect the internal structure of the real rock samples accurately. The physical experiment method is based on instruments such as FIB-SEM, Micro-CT, and Nano-CT to scan the rock samples and obtain grayscale images of the rock slices. Algorithms are used to stack the grayscale images to reconstruct the original structure of the rock samples into 3D digital core samples. This method can be divided into the sequential thin section imaging method, the confocal laser scanning method, and the CT scanning imaging method [5–7]. The CT scanning imaging method for rock cores is based on X-ray irradiation of the sample, where the differential absorption of X-rays by the various components of the sample generates varying levels of grayscale values in the resulting image. High-precision CT images are employed to establish a 3D digital rock core of the sample, using algorithms or software to reconstruct the true 3D morphology of the rock core. Currently, the principal techniques used in the field of digital rock core imaging are Micro-CT and Nano-CT. Dunsmuir first applied Micro-CT to the detection of rocks, ultimately obtaining three-dimensional images of rocks with pore-level resolution [8]. Coenen et al. constructed a micro-scale three-dimensional digital rock core using Micro-CT and analyzed the physical properties of carbonate rock cores [9]. Bin et al. established a multi-scale digital rock core using Micro-CT and Nano-CT, characterizing the distribution and structure of micro-pore throats in tight dune sand reservoirs of the Triassic Chang 7 Formation in the Ordos Basin [10]. Zhang et al. studied the characterization and evaluation methods of cracks and pores under the dual action of diagenetic compaction and tectonic compression in ultra-deep reservoirs. Zhou characterized coal rock samples using X-ray Micro-CT and fractal theory [11]. The pore-crack network structure was evaluated, and a simplified Sierpinski-style fractal model was proposed to describe the evolution process of the pore-crack network. Based on this, the changes in pore, permeability, and volume fractal dimension with axial stress were evaluated [12].

For the permeability prediction, the direct solution of the Navier–Stokes equations, the Pore network model [13], the Finite volume method [14], and the lattice Boltzmann method are employed to simulate the flow [15]. The pore network model (PNM) is a numerical tool to explore fluid flow and transport in porous media [16,17]. The pore network model has several advantages over traditional continuum models. It can not only handle complex geometries and topologies, but also capture the effects of pore-scale heterogeneity and anisotropy. In addition, the PNM can simulate transport processes over a wide range of length scales, from the molecular to the Darcy scale [18]. The PNM has been applied to a wide range of problems including flow porous media, transport in fuel cells, and diffusion in biological tissues. Moreover, it has also been used to study the effects of pore-scale heterogeneity on fluid flow and transport properties [19–21]. The lattice Boltzmann method (LBM) is a numerical simulation technique employed to study fluid dynamics and other complex systems, which is capable of simulating a wide range of fluid flow problems including those involving complex geometries and multiphase flows [22,23]. The lattice Boltzmann method has several advantages over traditional computational fluid dynamics methods. It is relatively easy to implement, and it is highly parallelizable, which makes it well-suited for large-scale simulations. In addition, the LBM can handle complex geometries without the need for complex mesh generation, and it is well-suited for simulating multiphase flows [24]. The lattice Boltzmann method has been applied to a wide range of problems including fluid flow in porous media, microfluidics, and

turbulence [25–28]. Although penetration rate prediction methods are currently mature, there remains a problem of high computational cost when calculating rock density for dense rocks.

In this study, we propose two low-cost methods for calculating the permeability of digital rock cores based on 3D digital cores, namely image resampling and image segmentation. Firstly, we obtained original grayscale images of homogeneous dune sand and fractured shale using Micro-CT scanning technology and constructed 3D digital cores for each lithology. We then applied image segmentation and image resampling to the CT images, calculated the permeability of the digital rock cores, and analyzed the adaptability of the two methods to the two rock samples. Our results demonstrated that image cutting and image resampling significantly reduce the computational cost, and the image cutting method can be applied to almost all rock samples.

## 2. Methodology

### 2.1. Shale and Dune Sand Samples

The fractured shale samples were obtained from Well Fan Ye 1 in the Dongying Sag of the Shengli Oilfield. The tectonic features of this oilfield block are characterized as laminated, layered, and blocky structures. The lithology is mainly oil shale, oil mudstone, and limestone. The shale is laminated, fragile, and contains numerous interlayer micrometer-sized fractures [29]. We used micron CT to scan the cores. The scanning resolution of the shale core is 6.5 μm, and that of the dune sand core is 8 μm. The shale sample measured porosity is 0.07 and permeability is 0.025 μm². Homogeneous dune sand samples were selected for comparison with the fractured shale samples to verify the applicability of the methods presented in this manuscript. The dune sand sample measured porosity is 0.32 and permeability is 1.41 μm².

### 2.2. Image Processing and Segmentation

To investigate the seepage parameters of the core in greater detail, it is necessary to perform image processing and segmentation on the original gray-scale images to construct a binary 3D digital core [10,11]. We extracted a representative region of interest within the core to reduce the computational cost associated with estimating the seepage parameters. The subvolume of the fractured shale sample is $512 \times 512 \times 512$ voxels, while that of the dune sand sample is $500 \times 500 \times 500$ voxels (Figure 1), and the grey part represents pore space or fracture, whereas the white part represents solid matrix/rock. During threshold segmentation, we manually adjusted the grayscale threshold until the pore space was completely filled (Figure 2). From the resulting binary digital rocks, we were able to calculate parameters such as porosity and permeability for fluid flow analysis.

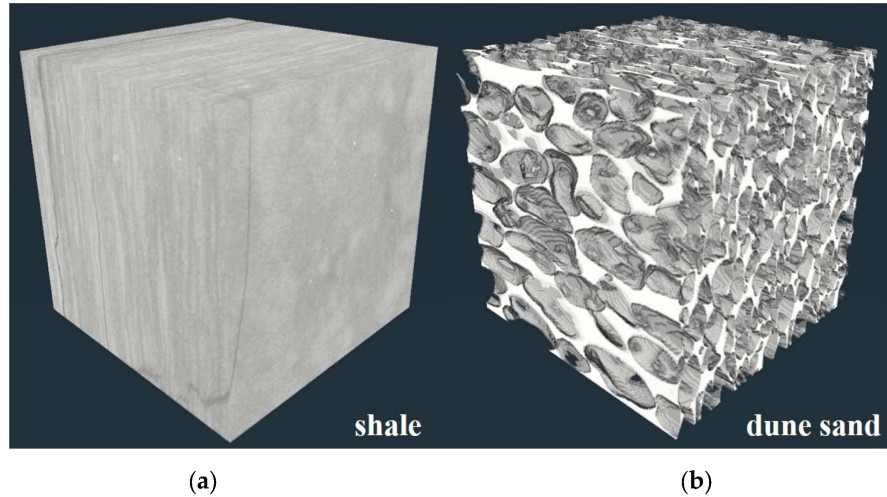

(**a**)　　　　　　　　　　　　　　　　　　　　　　　(**b**)

**Figure 1.** (**a**) Digital core of shale samples (**b**) Digital core of dune sand samples.

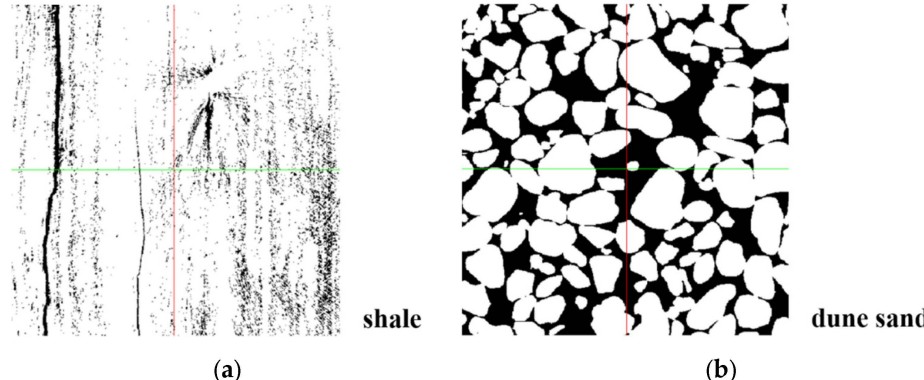

(a)                                       (b)

**Figure 2.** (**a**) Threshold segmentation of digital core of shale samples (**b**) Threshold segmentation of digital core of dune sand samples.

### 2.3. Image Resampling Method

Digital core image resampling is a technology that converts the original digital rock sample image into a low-resolution version. We implemented a resampling technique on binary digital cores to improve the efficiency of computing seepage parameters. For this purpose, we upscaled the binary digital core images obtained from threshold segmentation using the Lanczos interpolation algorithm. The resulting digital core images after resampling were then analyzed. The Lanczos filter uses a weighted sum of consecutive input samples as the interpolation value, allowing us to adjust parameters to balance computational speed and frequency response. This parameter also allows for a choice between smooth interpolation or sharp transients in the saved data. However, as with any such treatment, the boundary of the image cannot be accurately represented. The principle of Lanczos interpolation involves selecting a small window in the original image and calculating the weight to map it to a pixel in the new image [30]. The specific steps are the following:

The integer and floating-point coordinates of differing sampling points are calculated, with the weight information in the calculation template of the gray image. For the window templates corresponding to the Lancozs interpolation algorithm, the corresponding weight calculation formula for each position is the following:

$$L(x) = \begin{cases} \sin(x) \cdot \sin\left(\frac{x}{a}\right) & \text{if } -a < x < a \\ 0 & otherwise \end{cases} \tag{1}$$

Generally, $a$ is supposed to be 2 or 3, where the algorithm is suitable for image reduction interpolation ($a = 2$) or enlarged interpolation ($a = 3$).

The gray image of carbonate rock in this study adopts the Lancozs two-dimensional interpolation formula, and the weight is the product of two one-dimensional weight calculation results:

$$L(x, y) = L(x) \cdot L(y) \tag{2}$$

The weighted average of all pixels in the weight calculation window is the pixel value of the new pixel. According to the position of the input point, the weight $L(x)$ of different positions in the corresponding window is determined, and afterwards, the point values in the template are weighted and averaged. The formula is as follows:

$$S(x, y) = \sum_{i=|x|-a+1}^{|x|+a} \sum_{j=|y|-a+1}^{|y|+a} s_{ij} L(x-i) L(y-j) \tag{3}$$

In this study, we used the Lancozs interpolation algorithm with two upscaling factors (i.e., 2 and 4) to process the digital core images of shale and sand, obtaining the image of fractured shale with $256^3$, $128^3$ voxels (Figure 3) and that of dune sand with $250^3$, $125^3$

voxels (Figure 4) for subsequent porosity and permeability calculations. The blue part in Figures 3 and 4 represents pore space or fracture, whereas the grey part represents solid matrix/rock.

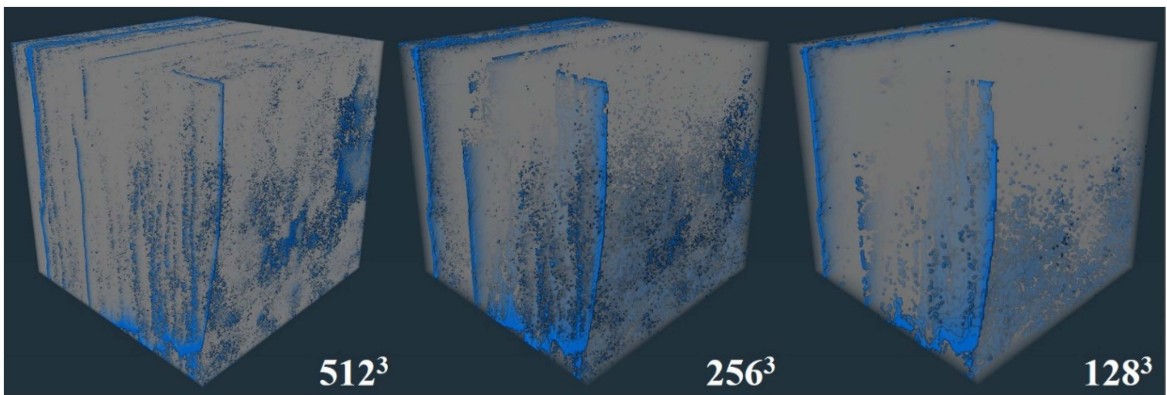

**Figure 3.** Digital core of fractured shale samples at different upscaling factors.

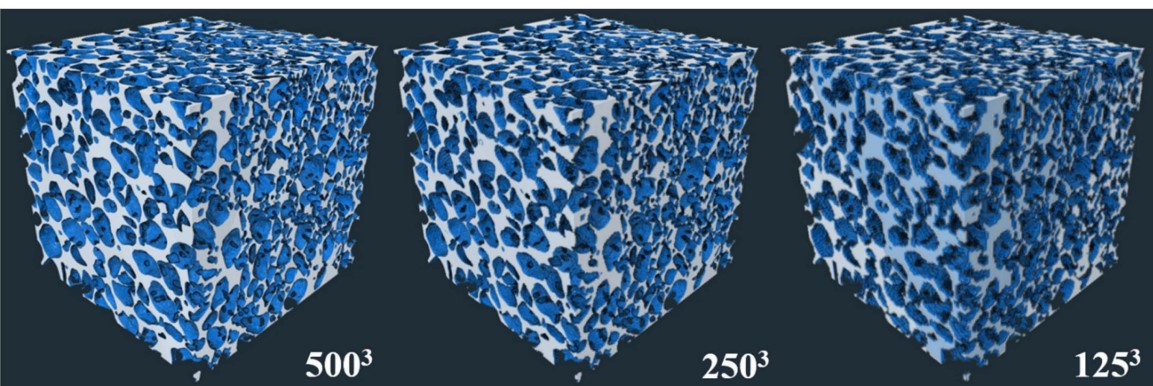

**Figure 4.** Digital core of fractured dune sand samples at different upscaling factors.

*2.4. Digital Core Image Cutting and Permeability Calculation*

Another low-cost method we used for permeability calculation is the digital core image cutting. We cut the binary digital core images obtained after thresholding into multiple subvolumes (Figure 5). For the fractured shale core images, we subdivided them into subvolumes of sizes $2 \times 2 \times 2$, $4 \times 4 \times 4$ and $8 \times 8 \times 8$, while for the dune sand core images, we constructed subvolumes of sizes $2 \times 2 \times 2$, $5 \times 5 \times 5$ and $10 \times 10 \times 10$. We computed the porosity and permeability of each subvolume in parallel, and then obtained the permeability of the original core using either the finite volume method or by fitting the porosity–permeability relationship.

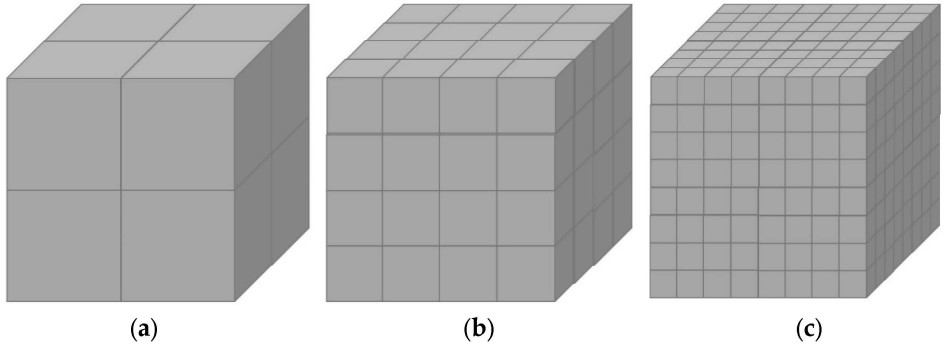

| (a) | (b) | (c) |

**Figure 5.** The digital core images were uniformly subdivided into (**a**) $2 \times 2 \times 2$, (**b**) $4 \times 4 \times 4$, and (**c**) $8 \times 8 \times 8$ subvolumes.

Here, we used a novel method for calculating permeability [31]. The traditional digital core permeability can be calculated by pore-scale simulation of the Stokes equation, but it tends to be time-consuming for dense rock samples. The calculation used in this paper is based on reducing the Stokes equation to the Darcy equation. Three-dimensional pore-scale simulation is simplified into multiple decoupled two-dimensional simulations, each of which provides a velocity distribution on a slice. Specifically, to simulate the two-dimensional fine-scale velocities solved by the Stokes equation, a local permeability is assigned to each voxel based on its velocity. According to this method, the three-dimensional Stokes equation can be simplified into multiple two-dimensional cases, providing local permeability distribution for the three-dimensional Darcy equation (an elliptic equation that is easy to solve by the algebraic multigrid method), thereby significantly reducing computational costs. The calculation error between the proposed method and LBM is found to be less than 10% on average in all tests. In this study, we also compared the two methods for the permeability of the dune sand sample, which validated our previous observation. Therefore, the permeability estimation by this method has little impact on the general conclusions. Considering that the proposed method has a significantly lower calculation cost, we adopt the proposed method to calculate the permeability of upscaled digital core images, as well as the permeability of each subvolume obtained after digital core image cutting in this work.

The finite volume method is a numerical technique used to solve fluid dynamics problems. It involves dividing the computational domain into small control volumes or cells, and then discretizing the governing equations over these control volumes using finite difference or finite volume methods. The resulting algebraic equations are solved iteratively using linear solvers. In this study, we consider the following elliptic equation that can be utilized in problems related to incompressible single-phase fluid flow [32].

$$\begin{cases} -\nabla \cdot (K(\boldsymbol{x})\nabla H(\boldsymbol{x})) = f(\boldsymbol{x}), & \boldsymbol{x} \in \Omega \\ c_1 H(\boldsymbol{x}) + c_2 \frac{\partial H(\boldsymbol{x})}{\partial n} = g(\boldsymbol{x}), & \boldsymbol{x} \in \partial\Omega \end{cases} \tag{4}$$

where $\boldsymbol{x} = (x, y, z)$, $\Omega \subset R^3$ is a cuboid domain with boundary $\partial\Omega$; $K$ is a symmetric positive-definite tensor, referred to as conductivity or permeability; and $H$ indicates the hydraulic head or pressure. Equation (4) is usually coupled with Darcy's law as $v = -K\nabla H$, called Darcy velocity. The equivalent conductivity for the coarse block is defined as [33]:

$$\frac{1}{V} \int_\Omega v(\boldsymbol{x}) d\boldsymbol{x} = K_{eq} \left( \frac{1}{V} \int_\Omega \nabla H(\boldsymbol{x}) d\boldsymbol{x} \right) \tag{5}$$

where $v$ is the total volume. Here, the fine-scale conductivity is diagonal [34]:

$$\boldsymbol{K} = \begin{pmatrix} Kx & & \\ & Ky & \\ & & Kz \end{pmatrix} \tag{6}$$

We solved the flow equation for each coarse grid block using a constant pressure boundary condition as (in 3D space):

$$\begin{array}{ll} P(y,z)|_{x=0} = 1 & P(y,z)|_{x=L_x} = 0 \\ v_y|_{y=0} = 0 & v_y|_{y=L_y} = 0 \\ v_z|_{z=0} = 0 & v_z|_{z=L_z} = 0 \end{array} \tag{7}$$

where $P$ represents the pressure at the boundary.

In this approach, we first computed the permeability of each subvolume obtained after digital core image cutting, and then calculated the permeability of the entire core using the finite volume method. We then compared the permeability calculated using the finite volume method with the permeability of the core to verify the feasibility of this method. Additionally, we fitted the porosity–permeability relationship for all subvolumes

and estimated the permeability of the core using the total porosity of the core. These two methods significantly reduce the computational cost of permeability calculation for digital cores.

## 3. Results

### 3.1. Digital Core Image Resample

By implementing the Lanczos interpolation algorithm for resampling binary digital rock images, we obtained digital rock images of shale and dune sand at different upscaling factors. Based on this, we calculated the porosity and permeability of rock images at different upscaling factors. According to Figure 6, as the upscaling factor increases, the porosity of dune sand hardly changes due to its low heterogeneity. Meanwhile, the permeability of dune sand in the $x$, $y$, and $z$ directions varies slightly with the upscaling factor, but the magnitude of the variation is small, with errors below 10%. Therefore, resampling of digital rock images can be used to roughly estimate the permeability of homogeneous rocks, which can significantly reduce the computational cost of permeability calculation. Figure 7 shows the change in permeability of fractured shale with upscaling factor. As the upscaling factor increases, the porosity of shale decreases, with most of the reduced porosity being non-connected micropores. Porosity and permeability vary significantly at different upscaling factors due to the strong heterogeneity of fractured shale. Therefore, resampling of digital rock images cannot be used to calculate the permeability of fractured shale. As shown in Figure 8, the method of resampling digital rock images can significantly reduce the computational cost of permeability calculation, but is solely applicable to relatively homogeneous rocks. Using image resampling to calculate permeability for highly heterogeneous rocks will result in significant errors.

### 3.2. Digital Core Image Cutting

By performing cutting processing on the digital core images of two types of rock samples, we uniformly subdivided the fractured shale core image into $2 \times 2 \times 2$, $4 \times 4 \times 4$, and $8 \times 8 \times 8$ subvolumes, and the homogeneous dune sand core image into $2 \times 2 \times 2$, $5 \times 5 \times 5$, and $10 \times 10 \times 10$ subvolumes. We used the computational-effective permeability calculation method (Liao et al., 2022) to calculate the permeability in the x, y, and z directions of each subvolume under constant pressure boundary conditions and obtained the porosity–permeability relationships for each subvolume. Figures 9–11 show the calculated permeability of all subvolumes of the medium (blue circle) and the overall permeability from different approaches for the dune sand sample. Subsequently, we fitted the log($\varphi$) and log(k) of these subvolumes using linear regression (black line). The fitting results for the dune sand sample are shown in Figures 9–11. The results indicate good fitting performance for the dune sand sample due to its relatively homogeneous lithology. Figures 12 and 13 present the fitting results for the fractured shale sample. As the y direction of the shale sample is perpendicular to the natural fractures and has a lower permeability, only the fitting results for log($\varphi$) and log(k) in the x and z directions are shown. Due to the small number of points fitted for the $2 \times 2 \times 2$ subvolumes of the shale sample, the fitting results are erroneous (Figure 12).

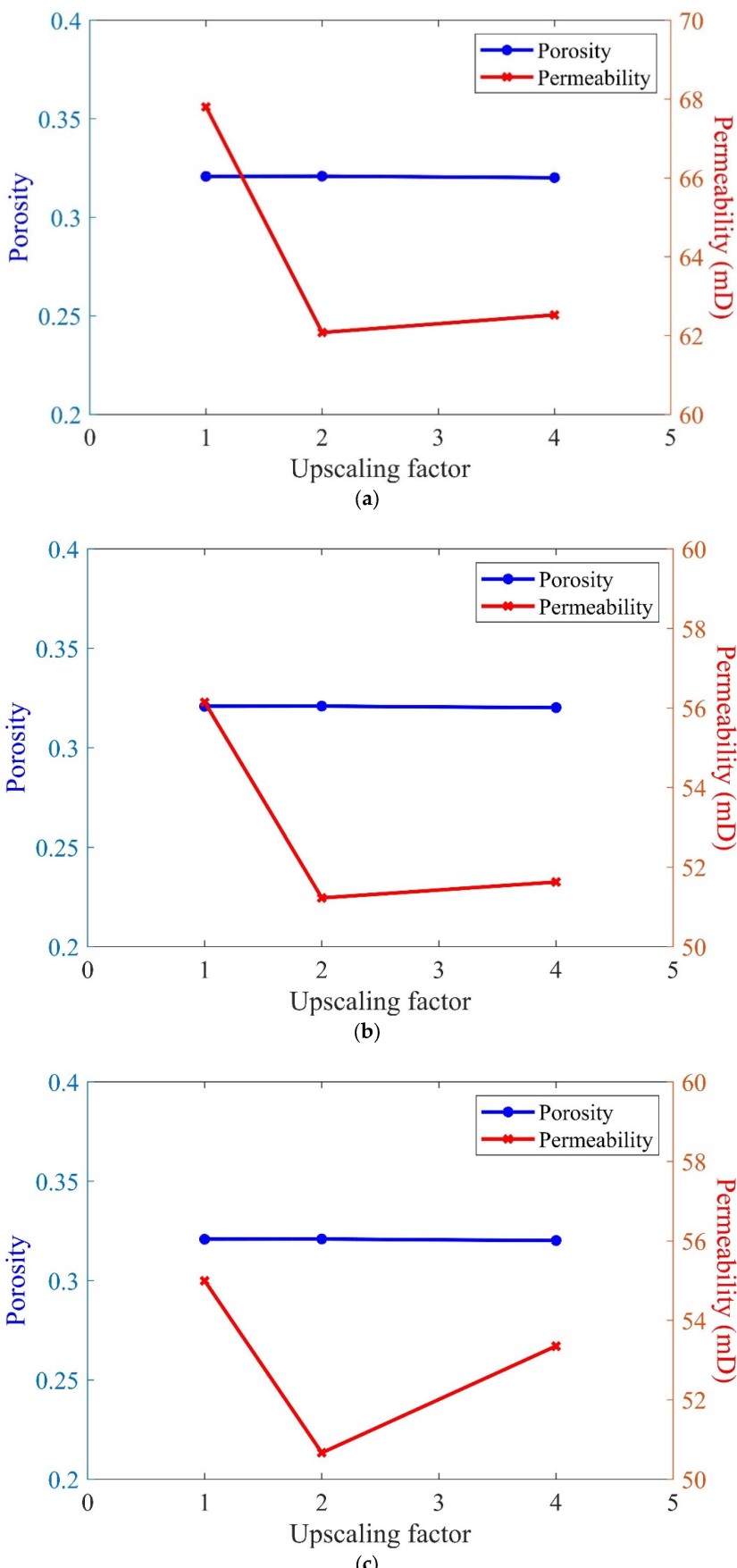

**Figure 6.** Variations of porosity and permeability of dune sand samples along the (**a**) *x*, (**b**) *y*, and (**c**) *z* directions with respect to upscaling factor.

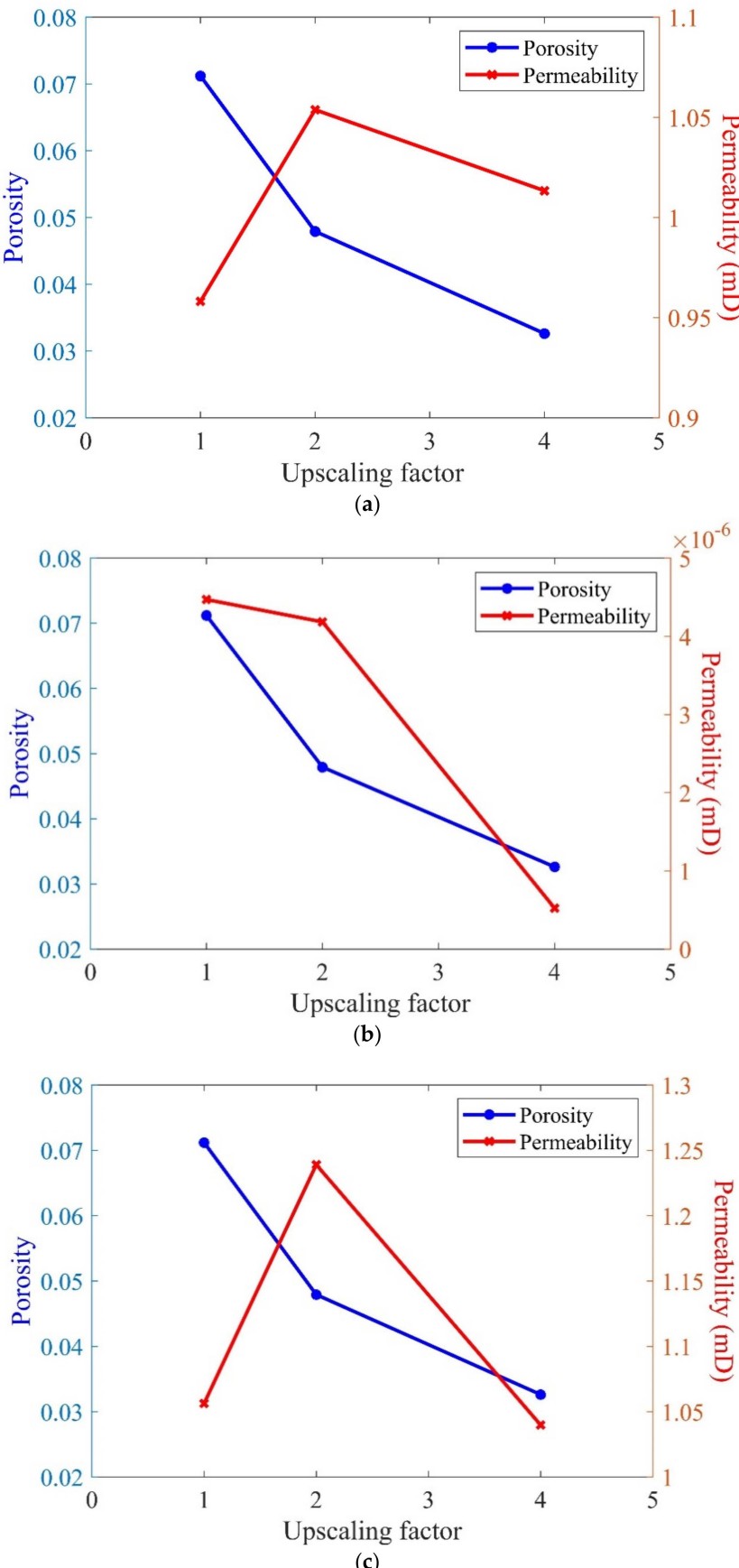

**Figure 7.** Variations of porosity and permeability of shale samples along the (**a**) *x*, (**b**) *y*, and (**c**) *z* directions with respect to upscaling factor.

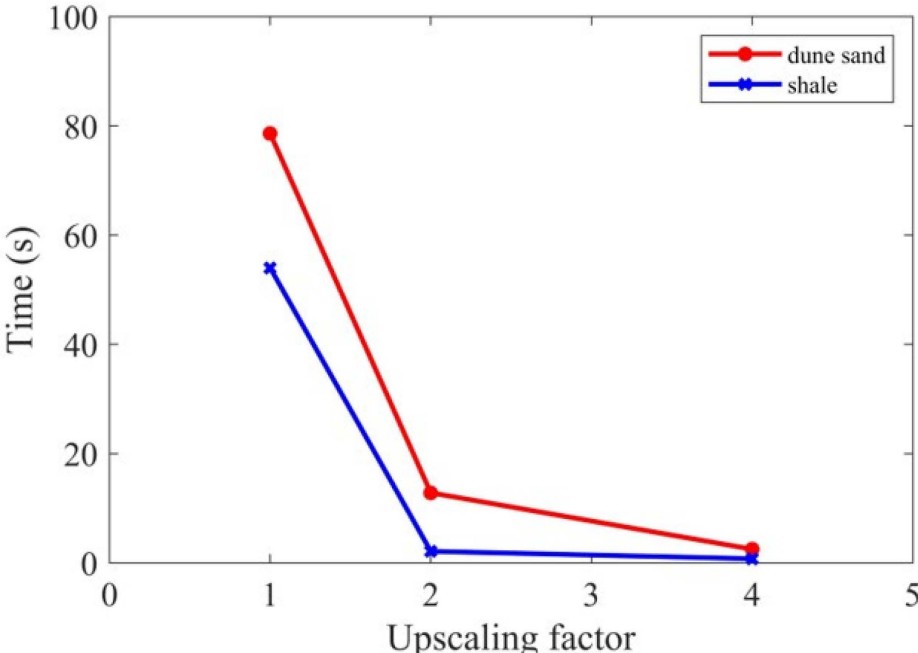

**Figure 8.** Variations of computation time for permeability calculation of digital core with respect to upscaling factor.

The relationship between porosity and permeability is usually expressed using the Kozeny–Carman equation [35]:

$$k = d^2 \frac{10^9}{72} \frac{\varphi^3}{\tau^2 (1-\varphi)^2} \tag{8}$$

where $k$ is in mD, $d$ is in mm, and the tortuosity $\tau$ is fixed at 2.5.

This equation is an empirical formula for predicting the hydraulic conductivity of porous media, which is advantageous in terms of simplicity and ease of use. The basic idea of the equation is that the permeability increases as the porosity grows and inversely proportional to the square of pore diameter. According to Figure 9, the permeability of the dune sand sample is proportional to the sixth or tenth power of porosity when the sample is uniformly subdivided into $2 \times 2 \times 2$ or $5 \times 5 \times 5$ subvolumes, which does not follow the Kozeny–Carman equation. In comparison, by uniformly subdividing the sample into $10 \times 10 \times 10$ subvolumes, the permeability is proportional to the cube of porosity, which shows good fitting results and is consistent with the Kozeny–Carman equation. This phenomenon indicates that when the sample is divided into enough number of subvolumes, the fitting results become more in line with the expected estimation. We substituted the porosity of the two rock types into the fitted equation and obtained the permeability results, which were compared with the original permeability of the rocks (Figures 9–13). The results demonstrate that the permeability obtained from the fitting of the homogeneous dune sand sample in the $x$, $y$, and $z$ directions is close to the original permeability, while the permeability obtained from the fitting of the fractured shale sample is significantly different from the original permeability due to strong heterogeneity.

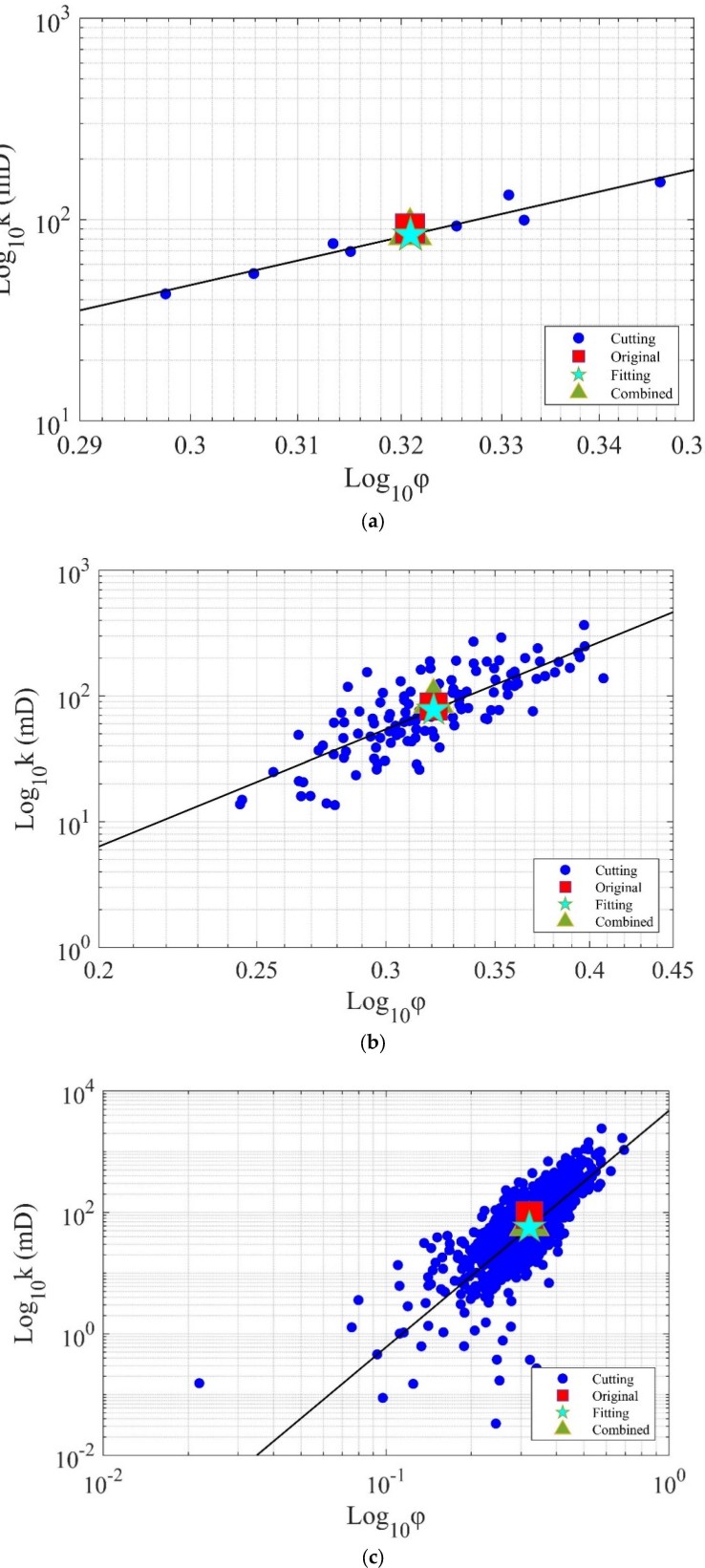

**Figure 9.** The relationship between porosity and permeability of dune sand digital core along the x direction was investigated after uniform subdivision into (**a**) 2 × 2 × 2, (**b**) 5 × 5 × 5, and (**c**) 10 × 10 × 10 subvolumes. Additionally, the permeability calculated using the finite volume method and the porosity–permeability relation fitting method was compared with the permeability of the core to validate the results.

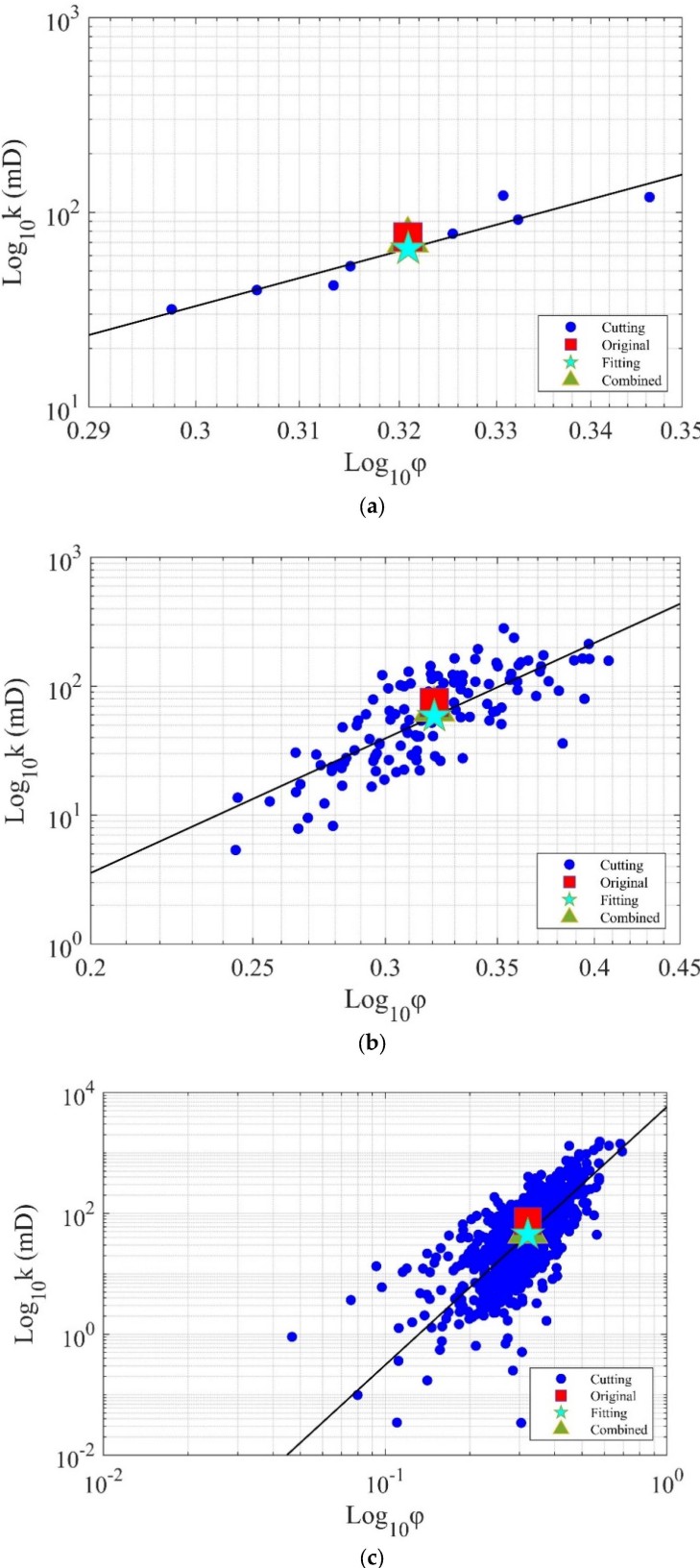

**Figure 10.** The relationship between porosity and permeability of dune sand digital core along the y direction was investigated after uniform subdivision into (**a**) 2 × 2 × 2, (**b**) 5 × 5 × 5, and (**c**) 10 × 10 × 10 subvolumes. Additionally, the permeability calculated using the finite volume method and the porosity–permeability relation fitting method was compared with the permeability of the core to validate the results.

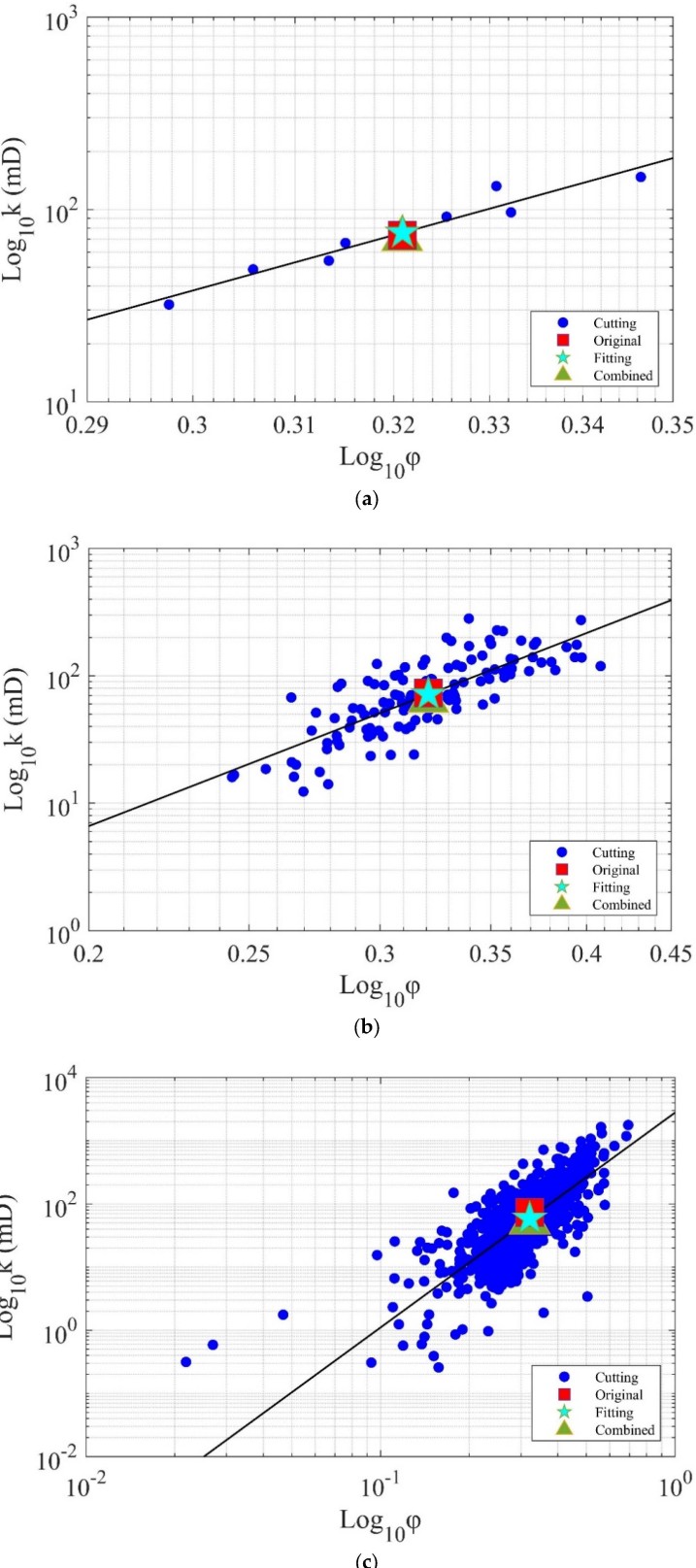

**Figure 11.** The relationship between porosity and permeability of dune sand digital core along the z direction was investigated after uniform subdivision into (**a**) 2 × 2 × 2, (**b**) 5 × 5 × 5, and (**c**) 10 × 10 × 10 subvolumes. Additionally, the permeability calculated using the finite volume method and the porosity–permeability relation fitting method was compared with the permeability of the core to validate the results.

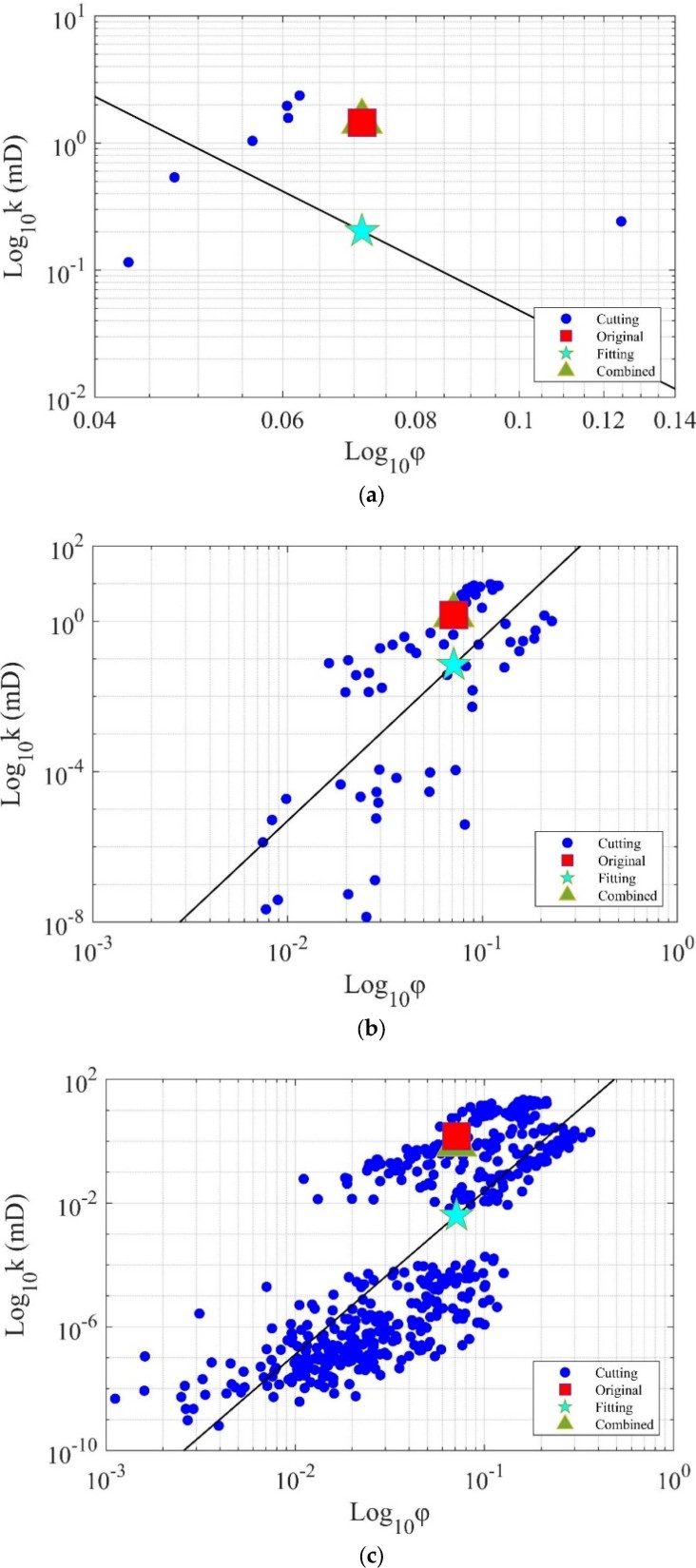

**Figure 12.** The relationship between porosity and permeability of shale digital core along the x direction was investigated after uniform subdivision into (**a**) 2 × 2 × 2, (**b**) 4 × 4 × 4, and (**c**) 8 × 8 × 8 subvolumes. Additionally, the permeability calculated using the finite volume method and the porosity–permeability relation fitting method was compared with the permeability of the core to validate the results.

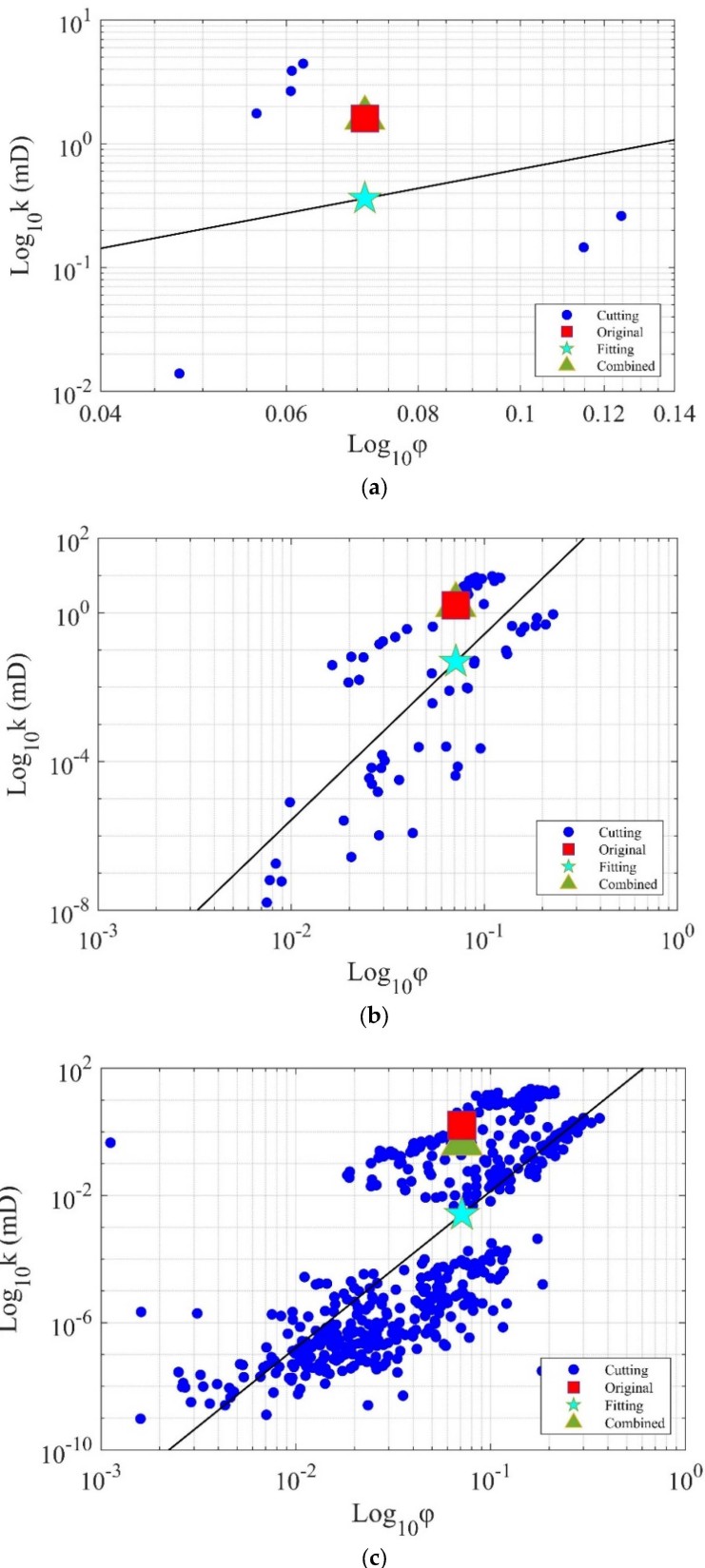

**Figure 13.** The relationship between porosity and permeability of shale digital core along the z direction was investigated after uniform subdivision into (**a**) 2 × 2 × 2, (**b**) 4 × 4 × 4, and (**c**) 8 × 8 × 8 subvolumes. Additionally, the permeability calculated using the finite volume method and the porosity–permeability relation fitting method was compared with the permeability of the core to validate the results.

In addition, we used the finite volume method to calculate the permeability of the entire rock core based on the subvolumes obtained by uniform subdivision, and compared the results with the original permeability of the rock samples (Figures 9–13). The results illustrate that the permeability calculated by the finite volume method for both types of rocks was close to the original permeability of the rock samples. Moreover, the smaller number of subvolumes generated by the uniform subdivision can achieve a higher accuracy of the calculation result. According to Figure 13, when the shale core was subdivided into $2 \times 2 \times 2$ subvolumes, the calculation error of the finite volume method was only 4.5%. Compared to the core subdivided into $10 \times 10 \times 10$ subvolumes, the calculation error reached 29%, which was due to the boundary effect. Figure 14 shows the computation time of permeability calculated by the finite volume method. When the shale sample was uniformly subdivided into $2 \times 2 \times 2$ and $4 \times 4 \times 4$ subvolumes, the computation cost was significantly reduced. Comparably, when the sample was uniformly subdivided into $8 \times 8 \times 8$ subvolumes, the improvement in computational efficiency is not that substantial. The rule for dune sand samples was similar to that for shale. Therefore, when using the finite volume method to calculate the permeability of the samples, the number of subdivisions should not be too large to ensure the accuracy of permeability calculation at high computational efficiency.

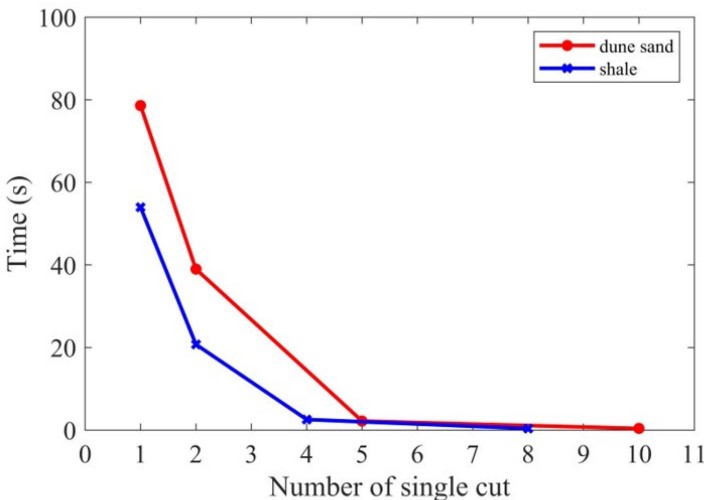

**Figure 14.** Variations of computation time for permeability calculation of digital core with respect to the number of single cuts.

Based on the above analysis, we summarize the applicable situations of different high-efficiency permeability calculation methods for different rock samples. For relatively homogeneous rock samples, three appropriate methods, namely digital rock image resampling, porosity–permeability relationship fitting, and digital rock image cutting, can be used to calculate permeability, which can improve the calculation efficiency of permeability while ensuring accuracy. For rock samples with strong heterogeneity, the digital rock image cutting method can be used to calculate permeability, which can greatly reduce the calculation cost of permeability.

## 4. Conclusions

In this study, we constructed 3D digital cores of fractured shale and homogeneous dune sand, and adopted two image processing methods, digital core image resampling and image cutting, to reduce the computational cost of digital core permeability. The image resamples method used lanczos interpolation algorithm to obtain binary digital core images with different upscaling factors, which would be used for subsequent porosity and permeability calculations. The image cutting method is to uniformly subdivide the binary digital core image into multiple subvolumes, and calculate the permeability using

either the finite volume simulation method or the porosity–permeability relationship fitting method. Then, we used an efficient permeability calculation method to reduce the complexity of permeability calculation. Especially, we summarized the applicability of different image processing methods for different rock samples, providing prerequisites for high computational cost digital core permeability calculation.

1. After using the Lanczos interpolation algorithm for image resampling of digital rocks, we obtained digital rock images with different upscaling factors. The changes in porosity and permeability of dune sand digital rock after image resampling were small, with an error of less than 10%. In contrast, the porosity of fractured shale decreased continuously with the increase of digital rock upscaling factor due to its strong heterogeneity. The decreased porosity usually corresponded to micro-pores or disconnected pores. The permeability of shale did not show an obvious trend with the increase of upscaling factor and had a significant calculation error.

2. After uniformly subdividing digital rock images into different numbers of subvolumes, the permeability of the rock can be calculated using the finite volume method or the porosity–permeability relationship fitting method. For the relatively homogeneous dune sand sample, both methods result in small errors in permeability calculation. For fractured shale, the finite volume method has a higher accuracy in permeability calculation, but the fitting method of the porosity–permeability relationship performs poorly due to its strong heterogeneity. Concerning boundary effects, the number of subdivisions should not be too large when using the finite volume method; otherwise, the calculated permeability will have an increased error.

3. The digital rock permeability calculation efficiency can be greatly improved by the image resampling methods and image cutting methods. However, these two methods are not applicable to all types of rocks. For relatively homogeneous rocks, all methods can be used to calculate the permeability of digital rocks with small errors. For strongly heterogeneous rocks, only the finite volume method can guarantee the accuracy of permeability calculation. In summary, the finite volume method can be applied to all types of rocks for digital rock permeability calculation, which can greatly improve the calculation efficiency while ensuring accuracy.

**Author Contributions:** Conceptualization, Q.L.; methodology, X.G.; software, S.Y.; validation, M.C.; formal analysis, Q.L.; investigation, M.C.; resources, M.S.A.; data curation, S.P.; writing—original draft preparation, S.Y.; writing—review and editing, M.S.A.; visualization, X.G.; supervision, S.P. All authors have read and agreed to the published version of the manuscript.

**Funding:** The authors would like to thank the support from the State Key Laboratory of Shale Oil and Gas Enrichment Mechanisms and Effective Development (No. 33550000-22-ZC0613-0272), and the Science Foundation of China University of Petroleum, Beijing (No. 2462022QNXZ002).

**Conflicts of Interest:** The authors declare no conflict of interest.

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
