# Peer review of "Digital Core Permeability Computation by Image Processing Techniques"

_water, doi:10.3390/w15111995_

Round 1

Reviewer 1 Report

This paper introduces the application of digital core image processing technology in rock permeability calculation, proposes two low-cost digital core image processing methods, and verifies the applicability of these methods. The article is interesting and can be accepted after minor revision. Here, I have a few suggestions for the authors' consideration:

1. For the digital core image resampling technology, based on the content in line 101 and Section 2.3, this paper only mentions the purpose and specific methods of using the technology, without giving a clear definition. It is recommended to add a sentence, such as "Digital core image resampling is a technology that converts the original digital rock sample image into a low-resolution version."

2. The icon sizes of the figures in the article should be unified, and it is recommended to confirm whether the annotations under the figures should be centered.

3. In line 210, v = −K∇H, the variable name 'v' should be in italics." I hope these revised comments help you provide clear feedback to the authors.

Reviewer 2 Report

This contribution presents a methodology for the calculation of the intrinsic permeability of porous media (of geologic origin) based on the structural properties of the materials at sub-pore-scale resolutions. The authors refer to this approach as 'digital core analysis'. While the specific approach is actually already established for calculating the REV properties of geologic porous media, this contribution presents some interesting results that would merit publication after some clarifications are provided. 

1. In the Methodology Section, the authors should report how the micro-CT scans of the porous domains are obtained, i.e. the type of the X-ray infrastructure, typical scan resolution and estimated resolution of typical pore for each class of materials.

2. In Fig. 1 it is not clear whether the grey color represents the solid material or the pore space. It appears that the pore space is transparent for the dune sand sample, while the opposite applies for the shale sample.

3. In the caption of Fig. 3, the authors should again clarify what the colors black and white represent.

4. The same should be applied in the captions of Fig 3 and 4. What do the blue, grey colors and the transparency represent? Solid or void space?

5. In Section 2.4 it is not clear which computational method is used to solve for the permeability of the binary subdomains. The authors argue that it is a novel method that simplifies the 3D Stokes flow to 2D Darcy flow in 2D slices of the digital porous media. As presented, this appears to be an over-simplification given that the whole idea behind 'digital pore physics' is to upscale by solving the conservation equations at the very fine pore scale to obtain accurate REV-scale properties, such as the Darcian permeability.

Therefore, any simplification at the pore-scale resolution (such as the one implied here) would render the method inaccurate.

6. Figs 9-11 appear to show the calculated permeability of each slice of the medium (blue dots) and the overall permeability of the domains vs the slice porosity. This is in fact an expected numerical result, given that the medium permeability and porosity is the volume-average of the permeabilities and porosities of the individual slices. Why is this an important result to show here?

7. Line 331. How is the tortuosity of the medium calculated to be equal to 2.5?

8. In the Introductory Section, the literature review of available pore-scale reconstruction should be extended to include the original contributions of

a) Quiblier, J. A. [1984]. ‘A new three-dimensional modeling technique for studying porous media.’ Journal of Colloid and Interface Science, 98(1), pp. 84–102

b) Adler, P., C. Jacquin & J. Quiblier [1990]. ‘Flow in simulated porous media.’ Interna- tional Journal of Multiphase Flow , 16(4),

c) Kainourgiakis et al (2005). Digitally Reconstructed Porous Media: Transport and Sorption Properties. Transport in Porous Media, 58, 43-62.pp. 691–712.edia.’ Journal of Colloid and Interface Science, 98

(1), pp. 84–102Quiblier, J. A.

[1984]. ‘A new three-dimensional modeling technique for Ka
